# The value of inquiring about functional impairments for early identification of inflammatory arthritis: a large cross-sectional derivation and validation study from the Netherlands

Bastiaan T van Dijk [1], Hanna W van Steenbergen,[1] Ellis Niemantsverdriet,[1] Elisabeth Brouwer,[2] Annette HM van der Helm-van Mil [1,3]

► Prepublication history and additional materials for this paper is available online. To view these files, please visit the journal online (http://dx.doi.org/10.1136/bmjopen-2020-040148).

[1]Rheumatology, Leiden University Medical Center, Leiden, The Netherlands
[2]Rheumatology and Clinical Immunology, University Medical Centre Groningen, Groningen, The Netherlands
[3]Rheumatology, Erasmus Medical Center, Rotterdam, The Netherlands

**Correspondence to**
Dr Annette HM van der Helm-van Mil;
a.h.m.vanderhelm@erasmusmc.nl and
Dr Bastiaan T van Dijk;
B.T.van_Dijk@lumc.nl

## ABSTRACT

**Objectives** Healthcare professionals other than rheumatologists experience difficulties in detecting early inflammatory arthritis (IA) by joint examination. Self-reported symptoms are increasingly considered as helpful and could be incorporated in online tools to assist healthcare professionals, but first their discriminative ability must be assessed. As part of this effort, we evaluated whether inquiring about functional impairments could aid early IA identification.

**Design** Cross-sectional derivation and validation study.

**Setting** Data from two Early Arthritis Recognition Clinics (EARC) in the Netherlands were studied, which are easy access outpatient rheumatology clinics intermediary between primary and secondary care for patients in whom general practitioners suspect but are unsure about IA presence.

**Participants** Between 2010 and 2014, 997 patients consecutively visited the Leiden-EARC (derivation cohort). Patients consecutively visiting the Groningen EARC (2010–2014, n=506) and Leiden-EARC (2015–2018, n=557) served as validation cohorts.

**Primary and secondary outcome measures** Physical functioning was assessed with the Health Assessment Questionnaire Disability-Index (HAQ); IA presence by physical joint examination by rheumatologists. HAQ questions were studied individually regarding discriminative ability for IA presence. For the best discriminating question, ORs and positive predictive values (PPVs) for IA presence were determined.

**Results** IA was ascertained in 43% (derivation cohort), 53% and 35% (validation cohorts). In the derivation cohort, IA presence associated with higher mean HAQ scores (0.84 vs 0.73, p=0.003). One question on difficulties with dressing equalled discriminative ability of the total HAQ score. 'Difficulties with dressing' yielded ORs for IA presence of 1.8 (95% CI 1.4 to 2.4) in the derivation cohort; 2.0 (1.4 to 2.9) and 2.1 (1.5 to 3.1) in the validation cohorts. After adjustments for clinical characteristics these were 1.7 (1.3 to 2.3), 1.6 (1.1 to 2.5) and 1.9 (1.2 to 2.9). PPVs (probabilities of IA for positive answers) ranged 42%–60% and negative predictive values (probabilities of no IA for negative answers) ranged 57%–74%.

## Strengths and limitations of this study

► We evaluated the value of questions about functional impairments for early recognition of inflammatory arthritis (IA), for which new and simple tools are urgently needed — especially in primary care.

► This was studied in a setting intermediary between primary and secondary care that services patients whose general practitioners were doubtful about the presence of IA; the population for whom such new tools are most relevant.

► This association was tested for independence from known predictors and validated in two independent cohorts.

► However, further validation in primary care is needed, since data were not yet collected in actual primary care practices.

**Conclusions** Patient-reported difficulties with dressing in patients with suspected IA associated with actual IA presence. Although further validation is required, for example, in primary care, this simple question could be of help in future early IA detection tools for healthcare professionals with limited experience in joint examination.

## INTRODUCTION

In rheumatoid arthritis (RA), early initiation of treatment with disease-modifying antirheumatic drugs is associated with improved outcomes.[1 2] Also for other types of inflammatory arthritis (IA) the importance of early treatment is increasingly recognised.[3–5] To achieve early treatment initiation, however, it is necessary for at-risk patients to be referred to a rheumatologist in a timely fashion.[6]

Consequently, general practitioners (GPs) play a crucial role in early identification of IA, but they have indicated considerable barriers regarding timely and adequate referral of patients with musculoskeletal complaints,

including a lack of self-confidence in detecting synovitis.[7] GPs generally have limited experience with the detection of IA by joint palpation compared with rheumatologists, since their patient population is characterised by a relatively low incidence of RA and IA (0.7/1000 patient-years in the Netherlands, which amounts to 1–2 new patients on average per GP per year) compared with a much higher incidence of other musculoskeletal complaints.[8–10] These circumstances may add to referral delay at the level of the initial healthcare provider (usually the GP), which was found to be an important contributor to overall delay until treatment initiation in many European countries.[11]

In this light, the search for novel strategies to improve early recognition and referral of patients at risk for IA and RA remains of great importance. This was emphasised in the latest update of the European League against Rheumatism research agenda for early arthritis by including 'the identification of tools that could help GPs diagnose early arthritis and prioritise referral.'[6] Given the increasing usage of healthcare-related mobile applications (apps),[12] new tools are preferred to be simple and time-efficient, providing suitability for possible future digital initiatives for early IA screening. In addition to ease of use, such tools should be based on valid and reliable data but currently most medical mobile apps are not properly evidence based, highlighting the need for thorough research.[13]

Patients with IA generally experience limitations in physical functioning. One could hypothesise that these limitations may differ in sort and severity compared with those related to other, less urgent causes of joint symptoms. Functional impairments in patients with rheumatic diseases are generally assessed using the Health Assessment Questionnaire Disability Index (HAQ),[14] but use of this patient-reported outcome instrument is mostly limited to research in established rheumatic disease instead of earlier stages. Nonetheless, it is generally considered a self-explanatory and relatively efficient questionnaire (it can be completed within 5 minutes).[15] In order to search for tools that might facilitate early identification of IA we performed a cross-sectional study on the association between functional impairments (measured by the HAQ) and presence of IA at joint examination (identified by rheumatologists). Aiming for simple and time-efficient use in daily practice, we also studied the HAQ questions separately and evaluated whether a single question could facilitate early identification of IA. Although information on functional impairments alone is not expected to have sufficient accuracy, it may be used as part of a more comprehensive tool that could be incorporated in a screening app for early IA. Findings were verified in two validation cohorts.

## PATIENTS AND METHODS
### Patients
Aiming to decrease referral delay in early IA, Early Arthritis Recognition Clinics (EARC) were initiated in 2010 at the Leiden and Groningen University Medical Centres in the Netherlands. The EARC design has been described previously.[16] In short, EARCs are clinics for patients in whom their GP suspects but is unsure about the presence of IA. These easy-access clinics are characterised by their unique intermediate setting, namely in between primary and secondary care. The EARCs have no waiting lists, can be visited by referred patients once to twice a week without an appointment and have markedly reduced referral delay.[16] GPs in the region are instructed to quickly refer any patient for whom they are unsure about the presence of IA, instead of applying a strategy of 'watchful waiting' or performing additional diagnostic tests. Notably, autoantibodies or other additional investigations are generally not performed by GPs in our region,[17] which is in accordance with the Dutch national guidelines.[18]

At the EARC, patients completed a brief clinical questionnaire on their symptoms, followed by a short consultation and full 66-joint examination by an experienced rheumatologist. If IA was detected at physical examination (by palpable joint swelling), patients were analysed at the regular rheumatology outpatient clinic for further diagnosis. Otherwise, they were referred back to their GP. Laboratory tests and imaging were not performed at the EARC.

The derivation cohort comprised 997 patients who consecutively visited the Leiden-EARC between September 2010 and April 2014. The first validation cohort consisted of 506 patients who visited the Groningen-EARC from September 2010 to January 2014. The second validation cohort consisted of 557 patients who visited the Leiden-EARC from November 2015 to December 2018. Patients who visited the Leiden-EARC from April 2014 to November 2015 did not fill out questions on physical functioning and were not included in this study.

### Patient and public involvement
Patient research partners agreed with the pathway of care at EARC and provided feedback on the questionnaire, which was expanded in 2012 with two questions.

### Health Assessment Questionnaire
Physical functioning was assessed during the same EARC-visit using the HAQ, a questionnaire containing 20 questions on impairments across eight functional categories.[14 15] Each question was scored by patients on a 4-point scale ranging 0–3, representing the degree of difficulties experienced when performing the corresponding activity, with 0 indicating no difficulties and 3 indicating full disability. The total HAQ score is calculated as the average across eight categories and ranges 0–3. Patients in the derivation cohort and patients in the first validation cohort filled out the full HAQ. Based on initial results from the derivation cohort, patients in the second validation cohort were asked to complete only question 1A on difficulties with dressing.

### Outcome
The outcome used in all statistical analyses was presence of IA, determined by the rheumatologist at physical examination. Final classifying diagnoses were made during

van Dijk BT, *et al. BMJ Open* 2020;**10**:e040148. doi:10.1136/bmjopen-2020-040148

subsequent visit(s) at the regular rheumatology outpatient clinic and were beyond the scope of this study. The rheumatologists who determined IA at joint examination were not formally blinded to the HAQ data; at request, however, the participating rheumatologists answered not to consider HAQ data in their evaluation of the presence of IA (personal communication).

## Statistical analyses

In the derivation cohort, total HAQ scores were compared between patients with and without IA at joint examination. Areas under the curve (AUCs) were determined for the total HAQ score as well as its questions individually, in order to select the best discriminating question for further investigation (based on highest AUC). The use of aids and devices was taken into account for calculation of the total HAQ score as prescribed, but not for scores on individual questions.[14 15] For the best discriminating question, ORs and AUCs were determined with both the original, categorical responses and with binary responses (no vs any degree of difficulties; i.e., scores ≥1), since a binary tool would be even more suited for easy use in daily practice. ORs were adjusted for age and gender.[19–21] Test characteristics and probabilities of IA were calculated. Analyses were repeated in the two validation cohorts.

A previous study from the Leiden-EARC derived an algorithm for IA identification, consisting of clinical variables that were independently associated with the presence of IA (being male, age ≥60 years, symptom duration, acuteness of symptom onset, morning stiffness >60 min, number of painful joints, presence of patient-reported swollen joint(s) and difficulties with making a fist).[21] These variables were included in multivariable logistic regression analyses, to study if self-reported functional disabilities were independently associated with IA. In addition, change in diagnostic accuracy when adding functional disability to the algorithm was assessed in the derivation cohort by comparing AUCs.

The HAQ question of interest was missing in 4%, 2% and 3% of the cohorts, respectively. Characteristics of patients with available and missing data on this HAQ question were compared. Missing values were imputed using multiple imputation by chained equations with predictive mean matching.[22] Rubin's rules were applied to pool point estimates and CIs across 30 imputations.[23] Imputed data were used for the main analyses. In sensitivity analyses, results were compared with those obtained from the unimputed data.

IBM SPSS (V.25) and R (V.3.5.2) were used. P values<0.05 were considered statistically significant.

## RESULTS

### Patients

Patient characteristics for all three cohorts are presented in table 1. Over 80% in each cohort presented with hand and/or wrist arthralgia. IA was diagnosed in 43%, 53% and 35% of patients, respectively. Patients in the first validation cohort had a slightly longer symptom duration

**Table 1** Characteristics of patients visiting the Early Arthritis Recognition Clinics

| | Derivation cohort (n=997) | First validation cohort (n=506) | Second validation cohort (n=557) |
|---|---|---|---|
| Inclusion period, years | 2010–2014 | 2010–2014 | 2015–2018 |
| Male, n (%) | 290 (29) | 177 (35) | 183 (33) |
| Age in years, mean±SD | 51±16 | 52±16 | 54±16 |
| Symptom duration in weeks, median (IQR) | 12 (4–62) | 18 (5–69) | 13 (4–57) |
| Acute onset of symptoms, n (%) | 368 (38) | 187 (38) | 203 (38) |
| Morning stiffness in minutes, median (IQR) | 10 (0–30) | 10 (0–31) | 10 (0–30) |
| No. of painful joints, median (IQR) | 7 (3–15) | 9 (4–18) | 8 (2–17) |
| No. of patient-reported swollen joints, median (IQR) | 2 (1–5) | 3 (1–8) | 2 (1–5) |
| Difficulties with making a fist, n (%) | 242 (47) | 130 (54) | 291 (54) |
| Arthralgia location, n (%) | | | |
| ≥1 hand or wrist joint | 810 (81) | 424 (84) | 455 (82) |
| ≥1 forefoot joint | 254 (25) | 142 (28) | 152 (27) |
| ≥1 hand, wrist or forefoot joint | 869 (87) | 454 (90) | 486 (87) |
| Exclusively large joints* | 22 (2) | 18 (4) | 9 (2) |
| HAQ score, mean±SD | 0.77±0.59 | 0.88±0.67 | −† |
| IA present at joint examination, n (%) | 426 (43) | 267 (53) | 193 (35) |

Original, unimputed data.
*Shoulder, elbow, hip, knee or ankle.
†Patients in the second validation cohort did not complete the full HAQ but only the question on difficulties with dressing.
HAQ, Health Assessment Questionnaire; IA, inflammatory arthritis.

### A. Total HAQ

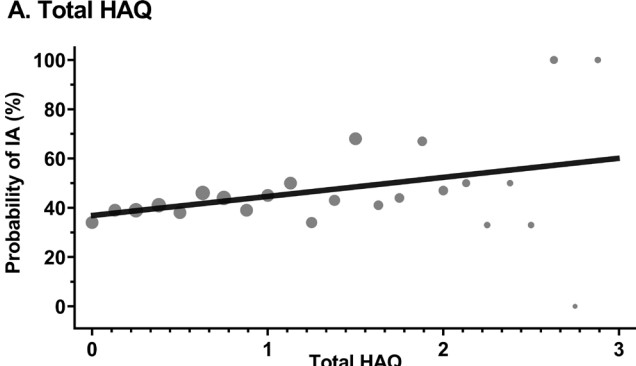

### B. Difficulties with dressing (original categories)

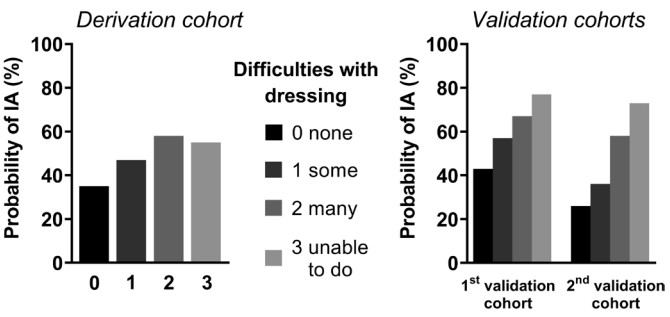

### C. Difficulties with dressing (dichotomised)

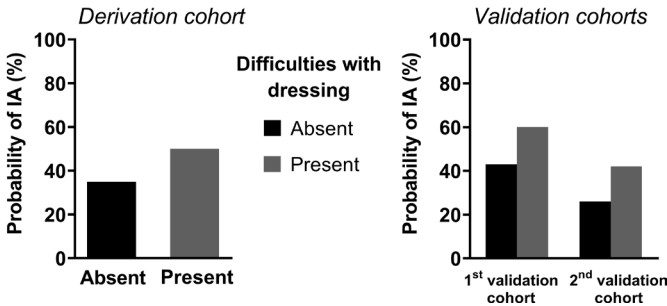

**Figure 1** Observed probabilities of IA according to (A) the total HAQ; (B) the degree of difficulties with dressing and (C) the presence of difficulties with dressing.
(A) Dot sizes vary according to the number of patients with that particular total HAQ score. The fit line was calculated by means of binary logistic regression. HAQ, Health Assessment Questionnaire Disability Index; IA, inflammatory arthritis.

**Table 2** Association between presence of IA and the degree of difficulties with dressing in the derivation cohort (n=997), with and without dichotomisation into binary scores

| Difficulties with dressing | | IA N (%*) | No IA N (%*) | OR (95% CI) | AUC (95% CI) |
|---|---|---|---|---|---|
| Categorical† | 0 | 169 (40) | 310 (54) | 1.00 (ref) | 0.58 (0.55 to 0.62) |
| | 1 | 188 (44) | 210 (37) | 1.64 (1.24 to 2.16) | |
| | 2 | 62 (15) | 45 (8) | 2.49 (1.61 to 3.84) | |
| | 3 | 7 (2) | 6 (1) | 2.20 (0.73 to 6.66) | |
| Binary | No (0) | 169 (40) | 310 (54) | 1.00 (ref) | 0.57 (0.54 to 0.61) |
| | Yes (≥1) | 257 (60) | 261 (46) | 1.80 (1.39 to 2.33) | |

Results were pooled across 30 imputations; percentages and numbers were rounded.
*Percentage of the total number of patients with and without IA, respectively.
†0='without any difficulty'; 1='with some difficulty'; 2='with much difficulty'; 3='unable to do'.
AUC, area under the receiver operating characteristics curve; IA, inflammatory arthritis; ref, reference.

To search for a method that is more simple and time-efficient for usage in daily practice than the 20-question total HAQ, its questions were also studied individually (online supplemental table S1). Of these questions, scores on 'difficulties with dressing' (question 1A) yielded the highest AUC, which equalled the AUC of the total HAQ score: 0.58 (0.55–0.62) vs 0.55 (0.52–0.59), respectively. This indicates that, in addition to greater ease of use, the discriminative ability of just one question was equal to the total HAQ. For these reasons, this question ('Are you able to dress yourself, including shoelaces and buttons?') was considered the best discriminating question and was studied further instead of the total HAQ score. The percentage of patients with different categories of impairment among the patients with and without IA are presented in table 2. The probabilities of IA presence for the different categories of impairments with dressing are indicated in figure 1B; the probability increased with more impairment.

Since a binary question is even easier to implement in practice, we assessed the discriminative ability of dichotomised scores on difficulties with dressing. Loss of overall discriminative ability, measured using the AUC, induced by dichotomisation into a binary score was minor (AUC of 0.57 (0.54–0.61) vs 0.58 (0.55–0.62)). Observed probabilities of IA presence with this dichotomised score are presented in figure 1C. For the benefit of ease of use this minor difference was accepted and dichotomised scores were used in subsequent analyses.

### Difficulties with dressing were independently associated with presence of IA in derivation cohort

ORs for the IA presence of a confirmative answer on the question about difficulties with dressing are presented in

(median 18 weeks) than patients in the derivation and second validation cohorts (median 12 and 13 weeks, respectively). All other characteristics were similar.

### Discriminative ability of a single question equalled the total HAQ score

Patients with IA at physical examination had a higher mean total HAQ score than patients without IA in the derivation cohort (0.84 (95% CI 0.77 to 0.89) compared with 0.73 (0.67 to 0.77), p=0.003), indicating a higher degree of functional disability. A gradual increase of the probability of IA at joint examination was observed as total HAQ scores rose (figure 1A).

**Table 3** Presence of difficulties with dressing associated with presence of IA independently from other clinical variables

| | OR (95% CI) univariable | OR (95% CI) age, gender adjusted | OR (95% CI) multivariable* |
|---|---|---|---|
| Derivation cohort (n=997) | 1.80 (1.39 to 2.33) | 1.83 (1.41 to 2.39) | 1.71 (1.27 to 2.32) |
| First validation cohort (n=506) | 2.00 (1.39 to 2.87) | 1.89 (1.31 to 2.72) | 1.64 (1.08 to 2.50) |
| Second validation cohort (n=557) | 2.14 (1.48 to 3.10) | 1.95 (1.32 to 2.86) | 1.87 (1.20 to 2.92) |

Results were pooled across 30 imputations.
*Adjusted for: being male, age ≥60 years, symptom duration, acuteness of symptom onset, morning stiffness >60 min, number of painful joints, presence of patient-reported swollen joint(s) and difficulties with making a fist (see online supplemental table S2 for details).
IA, inflammatory arthritis.

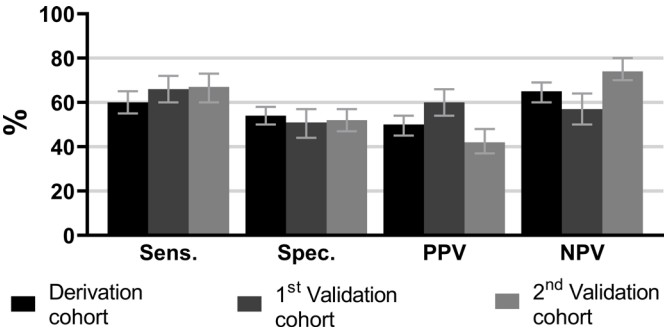

**Figure 2** Test characteristics of the presence of difficulties with dressing (scores ≥1), with IA at physical examination as outcome. Results were pooled across 30 imputations. Error bars indicate the 95% CI. IA, inflammatory arthritis; NPV, negative predictive value; PPV, positive predictive value; Sens., sensitivity; Spec., specificity.

table 3. In the derivation cohort, the presence of difficulties with dressing (scores ≥1) was associated with presence of IA with an age and gender adjusted OR of 1.83 (95% CI 1.41 to 2.39). After further adjustment for the aforementioned clinical variables that were previously identified as associated with IA (being male, age ≥60 years, symptom duration, acuteness of symptom onset, morning stiffness >60 min, number of painful joints, presence of patient-reported swollen joint(s) and difficulties with making a fist), difficulties with dressing remained independently associated with the presence of IA, with an OR of 1.71 (95% CI 1.27 to 2.32). ORs for the other clinical variables are provided in online supplemental table S2. The AUC of the model with only the aforementioned clinical variables was 0.72 (95% CI 0.69 to 0.76), while the model where difficulties with dressing was added had an AUC of 0.73 (95% CI 0.70 to 0.77). Since the presence of difficulties with dressing was independently associated with IA, it was subsequently studied in the validation cohorts.

### Difficulties with dressing were independently associated with presence of IA in both validation cohorts

In the first and second validation cohorts, age and gender adjusted ORs were 1.89 (95% CI 1.31 to 2.72) and 1.95 (95% CI 1.32 to 2.86), respectively. After further adjustment for the aforementioned clinical variables that were previously identified as associated with IA, ORs were 1.64 (95% CI 1.08 to 2.50) and 1.87 (95% CI 1.20 to 2.92), respectively. Thus, the independent association of difficulties with dressing with presence of IA was confirmed in both validation cohorts.

### Predictive values and test characteristics

Lastly, we studied test characteristics and predictive values of positive answers on just the single question about difficulties with dressing (figure 2). The negative predictive values (NPVs) in the different cohorts ranged between 57% and 74%, indicating that 57%–74% of patients without difficulties with dressing had no IA at

physical examination. The positive predictive values (PPVs) ranged between 42% and 60%, meaning that 42%–60% of patients who reported to have difficulties with dressing were indeed identified as having IA.

### Sensitivity analyses on unimputed data

No clinically relevant differences in characteristics of patients with and without information on 'difficulties with dressing' were found in online supplemental table S3. Subanalyses limited to the unimputed data (online supplemental table S4) yielded similar results as in the main analyses.

### DISCUSSION

Healthcare professionals who are not rheumatologists generally have limited experience in joint examination, which hampers them in early identification of IA. For GPs, this contributes to referral delay and necessitates the development of feasible tools that could support early identification of IA. We investigated the value of information on functional impairments in a cross-sectional study including patients with suspected but doubtful early IA according to GPs. We observed that, among patients with suspected IA according to GPs, those with actual IA according to the rheumatologist had higher HAQ scores, and that the discriminative ability of a binary answer to a single question on difficulties with dressing was equal to that of the total HAQ. Results were validated in independent cohorts of a similar setting, namely in between primary and secondary care. Together the results imply that easily obtainable information on functional impairments may be of value in facilitating early detection of IA.

The HAQ was originally designed for patient-reported outcome assessment and not as a diagnostic tool.[14] However, it is the most widely used and validated instrument in the field of rheumatology for measuring functional impairments.[15 24] The HAQ was neither designed to use in parts (and assess questions individually) when used as outcome measure. Despite these limitations, the current data suggest that a simple question on

difficulties with dressing, as deduced from the total HAQ, is of discriminative relevance. The simplicity of a single question and binary answers (no vs any difficulties with dressing) may facilitate the use of assessing functional impairments for diagnostic purposes in busy daily practice and as part of future internet or mobile app based tools for early identification of IA.

The best discriminating question was one related to hand function. Interestingly, over 80% of patients reported to have symptoms of the hands (table 1). Presumably, GPs are more often doubtful about IA in hand joints since these are relatively small joints and synovitis is often subtle. Alternatively, GPs may be more inclined to assess hand joints compared with other joints. Exploratory subgroup analyses limited to patients who indicated to have hand symptoms yielded similar results (data not shown).

The current study was, to the best of our knowledge, the first to investigate the discriminative ability of functional impairments measured by the HAQ in the context of identifying early IA. Several other tools to promote detection of early IA have been reported, including self-administered questionnaires and internet-based tools for primary care providers.[25–28] However, none of these were validated tools that incorporated patient-reported functional impairments. The best studied initiative, the Early Inflammatory Arthritis Detection Tool reported by Bell *et al*[25] contained one question on functional impairments in general, namely whether important activities were affected by joint problems. Although this tool showed good overall discriminative ability, it was externally validated only in a tertiary care setting.[29] Its usability in cases of diagnostic uncertainty remains undetermined.

It is unknown how patient-reported functional impairments currently influence referral policy in patients with suspected IA in primary care. Information on functional limitations is generally lacking in local guidelines and referral criteria.[18 30–36] The Dutch GP guideline on IA, for example, recommends asking patients about the impact of their joint complaints on their daily lives, but does not specify the information that should be collected and in which way this should be incorporated into decision making and referral policy.[18]

Although we attempted to seek for evidence on simple methods that are helpful to identify IA, the AUC reached by 'difficulties with dressing' was fairly limited. It is not surprising that a single question did not perform equally accurate as the reference, that is, evaluation of swollen joints with joint palpation. However, particularly in the current absence of other evidence-based strategies, this simple question that requires almost no extra effort, time or resources may be of value in improving early identification of IA.

The main strengths of the current study are related to the unique easy-access setting of the EA*R*Cs intermediary between primary and secondary care. First, the EARCs could be visited weekly without waiting list. Second, GPs in our regions were encouraged to refer any patient in whom they suspected but doubted about the presence of IA, resulting in a selection of patients in whom there is a lower and more uncertain suspicion of IA than would be the case at one of the more extensively studied Early Arthritis Clinics. The latter are secondary care facilities intended for patients with clear presence or a very high suspicion of IA. Other strengths are the validation of our findings in two cohorts and critical evaluation of the added clinical relevance by adjusting for previously reported diagnostic variables.

A limitation of our study is that data were not collected in primary care practices. Given the relatively low incidence of IA in primary care, it must be noted that prospective research on this subject in general practices would be particularly challenging and resource-intensive. Still, validation in primary care is the next step to be taken. Although the sensitivity and specificity are expected to remain stable across differences in prevalence of IA, PPV and NPVs are dependent on prior risks and may differ in primary care. Another limitation is that it is unknown how many patients were referred to but did not visit the EARC. Although we assumed this number to be relatively low because of the accessible nature of the EARCs, it may have influenced the prevalence of IA in the study population. Nevertheless, the prevalence observed in this study suggests that GPs in our region already perform quite well in prioritising referrals, since their suspicion of IA was correct in almost half of referred patients. This is in line with previous findings from the EARC.[16 21] Finally, as data were collected cross-sectionally and not longitudinally, analyses were not stratified for final diagnoses and patients who would develop RA were not studied separately. On the other hand, patients with other types of IA (eg, spondylo, psoriatic and undifferentiated arthritis) will most likely benefit from early treatment and referral to a rheumatologist as well.[3–5]

In conclusion, we studied the discriminative ability of functional limitations measured by the HAQ and individual HAQ questions in patients with suspected early IA. One question on difficulties with dressing by itself had a discriminative ability for actual presence of IA that equalled the full HAQ score and was validated in two cohorts. Although further research in primary care settings is necessary, the current data illustrate that a binary response to the question 'Are you able to dress yourself, including shoelaces and buttons?' is helpful in the assessment of patients with suspected early IA and could be used in future internet or mobile app based tools aiming for early identification of IA.

**Correction notice** This article has been corrected since it was first published. The middle initials for the first and last author has been added.

**Acknowledgements** We thank Xanthe M.E. Matthijssen for her statistical assistance.

**Contributors** BvD, HWvS and AvdH-vM designed the study. BvD, HWvS, EB and AvdH-vM contributed to data collection. All authors were involved in interpretation of data. BvD, EN and AvdH-vM drafted and edited the manuscript. All authors critically revised the manuscript.

**Funding**  This work was supported by the European Research Council (ERC) under the European Union's Horizon 2020 research and innovation programme (Starting grant, agreement No. 714312), and the Dutch Arthritis Society. These organisations were not involved in in the design, execution or reporting of this study.

**Competing interests**  EB as an employee of the UMCG received speaker fees and consulting fees from Roche in 2017 and 2018 which were paid to the UMCG (outside the submitted work). Otherwise non declared.

**Patient consent for publication**  Not required.

**Ethics approval**  The Leiden University Medical Centre (LUMC) medical ethical committee approved the study (P16.163) and granted a waiver for obtaining written informed consent in accordance with Dutch law on medical research due to data collection being limited to data acquired as part of usual care.

**Provenance and peer review**  Not commissioned; externally peer reviewed.

**Data availability statement**  Data are available from BvD (email: B.T.van_Dijk@lumc.nl) on reasonable request.

**ORCID iDs**
Bastiaan T van Dijk http://orcid.org/0000-0002-5161-6791
Annette HM van der Helm-van Mil http://orcid.org/0000-0001-8572-1437

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
