## [Reviewer comments · BMJ Open]

ARTICLE DETAILS

TITLE (PROVISIONAL)	The value of inquiring about functional impairments for early identification of inflammatory arthritis: a large cross-sectional derivation and validation study from the Netherlands
AUTHORS	van Dijk, Bastiaan; van Steenberg, Hanna; Niemantsverdriet, Ellis; Brouwer, Elisabeth; van der Helm-van Mil, Annette

VERSION 1 – REVIEW

REVIEWER	Carlo Alberto Scirè University of Ferrara Italy
REVIEW RETURNED	18-Jun-2020

GENERAL COMMENTS	I read with interest the paper entitled "The value of inquiring about functional impairments for early identification of inflammatory arthritis: a large cross-sectional derivation and validation study" by van Dijk, Bastiaan et al. This study addresses a relevant and unmet need of the early diagnosis of inflammatory arthritis. Unfortunately, all the evidence about the impact of early treatment in IA is far than fully translated into practice because of delays in referral. For this reason, the study is relevant. This study analyses the diagnostic properties of the HAQ score and specific subscores in the identification of IA patients among patients with suspected IA from the GP. The study design is appropriate, sample size adequate (even overpowered). High quality statistical methods were applied. I would just suggest improving the usability of the results, for example with diagnosis probability charts of different HAQ score values, or a formula to calculate individual probability of IA. In the present form, the results mainly suggest that difficulties in dressing increase the probability of IA. But clinicians are interested both in excluding (high sensitivity) or including (high specificity), not just about a relative increase of the risk. Having different cut offs for these two goals might be useful. Writing is easy to follow. I think that TRIPOD or STARD checklist are more appropriate for this study design.
---

REVIEWER	Titilola Falasinnu Stanford University School of Medicine
REVIEW RETURNED	23-Jun-2020

GENERAL COMMENTS	The manuscript presented by van Dijk et al evaluates the value of asking about functional impairments for early identification of IA.
---

	The authors validated their study in two independent population. They focused on the construct "having trouble getting dressed". However, this construct's PPV was only 50%. I would like the authors to take a step back and re-assess summing the HAQ questions to predict the presence of IA. This is because the use of only one measure is problematic and may lead to several missed cases. Were the rheumatologists blinded in any way?
--	---

VERSION 1 – AUTHOR RESPONSE

Reviewer 1: Carlo Alberto Scirè

– I read with interest the paper entitled “The value of inquiring about functional impairments for early identification of inflammatory arthritis: a large cross-sectional derivation and validation study” by van Dijk, Bastiaan et al.

This study addresses a relevant and unmet need of the early diagnosis of inflammatory arthritis. Unfortunately, all the evidence about the impact of early treatment in IA is far than fully translated into practice because of delays in referral.

For this reason, the study is relevant.

This study analyses the diagnostic properties of the HAQ score and specific subscores in the identification of IA patients among patients with suspected IA from the GP.

The study design is appropriate, sample size adequate (even overpowered).

High quality statistical methods were applied.

Answer: We thank the reviewer for all these positive comments.

– I would just suggest improving the usability of the results, for example with diagnosis probability charts of different HAQ score values, or a formula to calculate individual probability of IA. In the present form, the results mainly suggest that difficulties in dressing increase the probability of IA. But clinicians are interested both in excluding (high sensitivity) or including (high specificity), not just about a relative increase of the risk. Having different cut offs for these two goals might be useful.

Answer: We thank the reviewer for these sensible suggestions. The manuscript thus far mainly focused on situations wherein a higher sensitivity is preferred, since the detriments of missing IA-cases are expected to be higher than of extra referrals for joint examination. However, we do agree with the reviewer that opposite situations could be considered more as well. Two additions to the manuscript have been made accordingly:

- To improve usability, we prepared a figure that depicts the probability of IA according to varying HAQ scores (thus including low as well as high scores) on difficulties with dressing (new figure 1A).
- We agree that, next to diagnosis of IA, exclusion of IA is indeed important for individual patients as well. Therefore, negative predictive values were added to the results section of the abstract in order to put more emphasis on exclusion of IA.

In an attempt to specifically accommodate situations wherein a high specificity is preferred, we also explored test characteristics of the presence of many difficulties with dressing (scores ≥ 2 on the original categorical score) as opposed to no or some difficulties with dressing. This resulted in the following test characteristics in the figure below. Please compare these to the test characteristics achieved with the cut-off applied in the manuscript (scores ≥ 1 on the original categorical score). The increase in specificity was at the cost of a low sensitivity. In our opinion, due to the much lower sensitivity that the higher cut-off results in, this higher cut-off would unfortunately not be suitable for screening purposes.

Figure with scores ≥ 2 as cut-off:

Figure with scores ≥ 1 as cut-off, as presented in the manuscript:

– *Writing is easy to follow.*

Answer: We thank the reviewer for this positive comment.

– *I think that TRIPOD or STARD checklist are more appropriate for this study design.*

Answer: We thank the reviewer for pointing this out and fully agree. We will add the STARD checklist for diagnostic studies to the current submission, in addition to the STROBE checklist for observational studies.

Reviewer 2: Titilola Falasinnu

– *The manuscript presented by van Dijk et al evaluates the value of asking about functional impairments for early identification of IA. The authors validated their study in two independent populations. They focused on the construct "having trouble getting dressed". However, this construct's PPV was only 50%. I would like the authors to take a step back and re-assess summing the HAQ questions to predict the presence of IA. This is because the use of only one measure is problematic and may lead to several missed cases.*

Answer: We appreciate this comment of the reviewer, prompting us to provide more clarification regarding the choice to focus on one question instead of the full HAQ-score.

We understand the reviewer's concern and have now made a novel figure in which the probabilities for IA are plotted for the total HAQ (fig 1A), the single question in different categories (Fig 1B), and finally for the single question after dichotomisation (Fig 1C).

Although intuitively one may expect that a single question might be less predictive than the summed HAQ-score which incorporates more measures (namely 20 questions), we observed that the total HAQ did not perform better in detecting IA than just one of its questions (compare Fig 1A and 1B). This figure 1A shows that the probability of IA increases only very gradually with rising scores, especially considering the limited range of scores wherein the majority of participants fall. The discriminative ability (AUC) of the single question on difficulties with dressing was higher than that of the full HAQ, even in simplified dichotomised form.

We hope that this figure is helpful and insightful in depicting the data for the different steps.

We also adjusted the results section to better describe the steps that were taken and the data belonging to these steps. Thank you for pointing this out.

– *Were the rheumatologists blinded in any way?*

Answer: We thank the reviewer for pointing out this unclarity. Data, including the HAQ, were collected as part of routine medical care at the Early Arthritis Recognition Clinic (EARC). Therefore, formal blinding of the rheumatologist was not feasible. The method of outcome assessment however was independent from these data, namely by physical joint examination. Moreover, our rheumatologists indicated not to consider HAQ results when performing joint examination. We have now included this information in the results section.

VERSION 2 – REVIEW

REVIEWER	Carlo Alberto Scirè University of Ferrara
REVIEW RETURNED	11-Aug-2020

GENERAL COMMENTS	I have no further comments.
-----------------------------

REVIEWER	Titilola Falasinnu Stanford School of Medicine
REVIEW RETURNED	13-Aug-2020

GENERAL COMMENTS	Thanks for answering my questions
-----------------------------------

VERSION 2 – AUTHOR RESPONSE

Thank you very much for investing your precious time in reviewing this manuscript. Your approval, as well as your efforts which helped us to improve the manuscript, are greatly appreciated.